# IoT-Based Electricity Bill for Domestic Applications

**DOI:** 10.3390/s20216178

**Published:** 2020-10-29

**Authors:** Ramón Octavio Jiménez Betancourt, Juan Miguel González López, Emilio Barocio Espejo, Antonio Concha Sánchez, Efraín Villalvazo Laureano, Sergio Sandoval Pérez, Luis Contreras Aguilar

**Affiliations:** 1School of Electromechanical Engineering, Universidad de Colima, Manzanillo 28860, Mexico; jgonzalez71@ucol.mx (J.M.G.L.); villalvazo@ucol.mx (E.V.L.); 2Graduate Program in Electrical Engineering, Universidad de Guadalajara, Guadalajara 44430, Mexico; emilio.barocio@cucei.udg.mx; 3Faculty of Mechanical and Electrical Engineering, Universidad de Colima, Coquimatlán 28400, Mexico; aconcha@ucol.mx (A.C.S.); luisc@ucol.mx (L.C.A.); 4Tecnológico Nacional de México/Instituto Tecnológico de CD. Guzmán, Ciudad Guzmán 49100, Mexico; ssandoval@itcg.edu.mx

**Keywords:** smart grids, Internet of Things, home metering system, real-time electricity bill

## Abstract

This work proposes a real-time electricity bill for quantifying the energy used in domestic facilities in Mexico. This bill is a low-cost tool that takes advantage of the IoT technology for generating an easy reading real-time bill allowing the customers to constantly review and administrate their energy consumption. Using low-cost sensors and the electronic board Particle^®^ Photon, an energy meter is proposed. The presented prototype is extremely compact and satisfies safety measures to be used by anyone in a domestic installation. The measurement data is displayed and processed in real-time, and an appropriate algorithm determines the accumulated kWh. The energy consumed is displayed using an Html interface of easy interpretation for the customers, given recommendations about their consumption habits and some alarms in case of abnormal or high consumption. As a reinforcement measure for avoiding large consumption bills, the system is programmed to send messages to the user, remembering if the estimated consumption is large.

## 1. Introduction

The application of the technology for the Internet of Things (IoT) to monitor and control industrial systems has received a lot of attention around the world. In a very specific way, IoT systems have been gaining space in what is called intelligent electrical systems or Smart Grids [1]. The integration of IoT technology in intelligent electricity networks has the fundamental purpose of generating control alternatives to guarantee the efficiency and continuity of electrical energy systems. In the electrical industry, a great interest has arisen to determine the patterns at the domestic level, since with them, it is possible the administration of the energy, as well as the identification of failures in installations, identification of defective and/or obsolete appliances, and the generation of mathematical models related to the energy consumption. On the other hand, there is a strong tendency to incorporate this technology to be more efficient in the use of energy by the utilities. However, it is necessary that the end-users of energy at commercial and domestic levels also become aware and generate initiatives in their facilities to optimize the energy consumption [2].

In this context, household electricity users have always wondered if their energy consumption is that charged by the utility company. Initially, an option to respond to this concern is to place a kWh meter to validate the official meter reading. However, the answer to this first question originates other more complex questions, such as (1) Is the consumption due to energy leaks?; (2) Are the appliances consuming more energy than usual?; (3) How I can reduce the energy consumption?; and finally, (4) Does the consumption can be predicted? Nowadays, the Smart Home Energy Monitor system (SHEM) can response these questions. The main functions of the SHEM are to continuously read the energy consumption, to identify the appliances to be managed intelligently for energy savings, and to provide visual information of the energy consumed by the user [3]. The hardware needed for acquiring the signals can vary, but the core of the SHEM is composed by the processing and decision algorithms, and the visualization interfaces.

The SHEM can process the data consumption generated from various metering sockets installed in each of the appliances [4]. This scheme is known as intrusive load monitoring (ILM), and it permits the SHEM system to be more efficient since each appliance is monitored and controlled individually [5]. The main application appears when a complete home automation is needed, but it is usually composed by complex and expensive systems [5]. In counterpart, the non-intrusive load monitoring (NILM) scheme becomes the cheapest and the simplest option since only the measurement of total energy is needed [6]. The identification of habits of consumption becomes a more complex task, since all the appliances are aggregated to the global measurement and disaggregation methods should be used to identify the appliances in operation. This scheme can co-exist with home automation systems too, allowing the connection/disconnection of appliances. Regardless of the scheme used, the purpose of the SHEM is to generate information that allows the user to take decisions for an efficient energy consumption, as well as for diagnosis and prognosis as remedial actions to prevent large consumption in the future [7].

In the commercial aspect, a large of manufacturers of SHEM exist, which provide different models of different prices. Most of them have user-friendly monitoring interfaces. For example, some devices, such as the presented in Table 1, are becoming attractive non-intrusive options for home energy monitoring. The development of new algorithms and visualization interfaces is not possible since their architecture is closed.

In counterpart, there exist devices with an open architecture, which are developed by the company Openenergymonitor^®^ [8] that is a pioneer in this field. These devices offer various platforms for consumption monitoring such as emonPI^®^ or emonTx^®^. Moreover, they can be linked through the EmonCMS web application to store and process data in the cloud. Although, the devices presented in Table 1 have a slightly lower cost than $ 192.72 US, they do not include the web monitoring system, which must be purchased separately; however, they are an attractive option for researching purposes. Moreover, these devices allow implementing new algorithms, but their application to countries such as Mexico is limited due to their high cost and complex scheme of tariffs.

As previously mentioned, diagnosis and prognosis of consumption are inherent attributes in SHEM. These attributes are focus of research around the world. In this direction, several works have been presented with significant contributions. In this context, the prognosis of consumption from real measurements, for improving the use of heavy loads such heating systems, is studied in [9], where an Artificial Bee’s Colony algorithm (ABC) is proposed with a new metric called habitual average. The authors use a real database for test the ABC algorithm, and their results prove that the use of this metric as an internal variable helps to employ heating systems as an alternative for real implementation in the SHEM. This work is centered only in selecting intelligent decisions for efficient energy consumption, but functionalities to visualize results can be added to it.

The energy consumption behavior can be studied using industrial meters along with low-cost development boards. In this sense, reference [10] presents the utilization of an Echelon electricity meter for acquiring signals and a Raspberry board for their processing. The main goal in this work is to investigate the behavior of the consumer using data recovery mechanisms and machine learning methods. More recently, energy prediction has been studied in [11], where the multi-power state idea is introduced for classification and identification of appliances by a supervised learning. The proposed methodology in [11] shows a better prediction than that obtained with the binary power state model, and it represents an alternative for real-time home energy management. The work could be used with purposes of visualization too. On the other side, the massive application of IoT for smart homes is presented in [12], where the authors propose a system that uses the IoT and big data analytics tools for managing a large set of users. In this work, a Photon Particle^®^ board is used, and the authors present interfaces for the final user and the community owner. The interface to visualize the bill is not fully presented, but the results indicate that this board can easily be implemented on large scale smart home with billing purposes.

Recently, web-based processing systems are more common in forecasting due to the use of smart meter systems. The great variety of open source libraries on artificial intelligence allow to generate new ideas, such is the case of the work presented in [13], where a hybrid combination of SARIMA and metaheuristic firefly algorithm-based least squares support vector regression (MetaFA-LSSVR) is used for forecasting energy consumption. In addition, smart energy metering systems can be developed with low cost boards such as the presented in [14], where a LoRa-WiFi protocol is employed for reducing the dependency of internet access. This prototype allows detecting theft or abnormal consumptions through a cellular application. The system can be used for billing purposes, but this reference does not present a complete interface. Finally, the manufacturer industry has used development boards for diagnosis of energy consumption in their facilities [15]. In this case, they collect the measurement signals from industrial meters of the brand Schneider and Elmeasure, and by using a Raspberry board these signals are preprocessed. Interfaces for visualization of daily consumption patterns are generated for purposes of energy savings. As stated by the analysis of the previous works, note that the visualization of energy consumption of SHEM is a key aspect since the information should be displayed timely and easy interpretation. In function of these aspects the user can understand properly and take actions to save energy [16,17,18].

Motivated to this fact, this paper is focused on the development of an interface for real-time billing in the Mexico country, with the main objective of helping the users to avoid large energy consumptions and to save energy. The real-time bill is generated by a proposed IoT system that has the following characteristics: (1) it uses low-cost sensors and the electronic board Particle^®^ Photon; (2) it displays information about the consumption habits of the users; and (3) it activates alarms in case of abnormal or high consumption. The main contributions of this work are (i) the design of an interface that provides a real-time bill with the same characteristics of the bill generated by the company of energy, (ii) the forecast of energy consumption in a period using only one week of measurement, and (iii) the possibility to extend this measurement system for three-phase commercial applications.

This work is organized as follows. Section 2 describes how the energy in Mexico is charged, emphasizing mainly in the tariffs employed and the bill used by the company of energy. Additionally, the information presented in the bill is analyzed in detail, focusing on its interpretation and how this information can be correctly interpreted to help the user to save energy. On the other hand, Section 3 discuss the hardware constructed along with the algorithms and its validation. In Section 4, the interface designed is presented, and finally, Section 5 describes the application in a real situation.

## 2. Statement of the Problem

Commonly the energy bill emitted by the utilities around the world is a simple receipt indicating the consumed kWh in a monthly or bimonthly period, along with the total amount to pay and the rate applied [19,20,21,22,23,24,25,26]. Additional information such historical consumption during a year is displayed in order that the user can do a comparison between periods. Another less important information is about the total cost distributed between the different entities of the process from generation to delivery of the energy, but no more information is provided for the user.

In Mexico, the provided information in the bill by the Comision Federal de Electricidad (CFE) [27] is very similar to that mentioned in the last paragraph. However, the energy becomes cheaper when lower consumption is observed and expensive when a certain limit is reached (just three times the price per kWh), then the user can observe this information through a heat color bar in the printed receipt indicating the range of consumption and if it belongs to the expensive range. This information is not always understandable by the consumer; however, the company points out this aspect in the receipt by some messages such as: “to care off the energy” or “use of low energy equipment installation”; but not another more specific messages concerning the revision of the electrical installation or the deployment of a Photovoltaic system (PV).

Next, the main aspects regarding the energy bill are presented, starting with the analysis of the current tariffs and how this information is displayed in the printed bill, emphasizing how this information can be used properly in the designing of the real-time bill.

### 2.1. Energy Tariffs

There exist several domestic tariffs which depends mainly on the average minimum temperature in the summer where is applied. Additionally, to encourage the energy savings, the CFE has been adopted a block-rate tariff; that is, the lower the consumption the lower the cost per kWh. Table 2 summarizes the current tariffs and it emphasizes the ranges of consumption in a period [28].

Table 2 shows that some tariffs possess four blocks, and the range and costs can vary in each year and month. The actualized costs can be obtained from [27]. The Max level is estimated with the Simple Moving Average in a year for each region and if the user reaches this value before the end of year, automatically the price per kWh becomes block S independently of the kWh consumed (this tariff is well known as DAC). The user can access again to the initial tariff if the Max in the next year is not reached again. The geographic disposition of the tariffs around the country exhibits a complex distribution, and Figure 1 displays this information.

According to Figure 1, in some states coexists two or more tariffs, due to the wide variety of climates. Next section presents the information displayed in the current bill, which is used to design the proposed real-time bill.

### 2.2. CFE Energy Bill

The way to design the interface of the real-time bill was to review the information given by the CFE and how the user can use it efficiently. The images given in Figure 2 summarize the most relevant information for the user [27] (which corresponds to the translated version).

Figure 2a depicts the address and number of the user, the tariff, number of the meter, amount to pay in Mexican pesos, the period of consumption, the deadline to pay, and the day to cut the service in case of delay in the payment. As can be seen, tariff, period of consumption, total amount to pay, deadline, and the cut-off date are key information for the user. Figure 2b displays the kWh consumed through the initial and final readings of the meter in the period (445 kWh). From the consumed kWh, it is indicated how the corresponding tariff (1B) is applied, where the first 250 kWh (block B) has the lower price per kWh whilst the next 195 kWh (block IL) has a higher price per kWh. In this case, a consumption for block S is not observed; thus, the arrow over the color bar (in the middle) indicates that the consumption in this period is in the range of “moderate consumption”. This information is fuzzy and does not offer help to common users to avoid large consumption bills, thus limiting their possibility to implement countermeasures, such as to review the wiring of their installation, change of habits of consumption, installation of PV systems, among others. This information could be valuable if the user could know the time when the energy consumption has high cost. Finally, Figure 2c displays less important information of the total costs including state and federal taxes.

### 2.3. Bill Desingning

Following the description of the printed bill version, it can be concluded that the instantaneously reading of the accumulated kWh can be used for establishing the different strategies of visualization of the main information such as the kWh in each range, costs, charts for each day of the week and period, and recommendations when higher consumptions are observed. Additionally, information can be obtained and displayed using an appropriate visualization method to provide different alternatives to use the energy efficiently (for example, the forecast of the bill can be obtained with the moving mean average of the kWh). With this aim in mind, a web interface with a menu with different screens, that shows the most relevant information, is proposed. The objective of each screen of the interface is summarized in Table 3.

Each screen should display the most important information of the bill and appropriate strategies will be used. Additionally, some alerts via SMS could be provided. Table 4 describes the method used for the main variables of the interface.

## 3. The IoT-Based Measurement System

For the measurement of the energy consumed, we propose the use of an own designed non- intrusive system with compact characteristics and easy installation. The schematic of Figure 3 shows the underlying idea behind the placement of the measurement system in the electrical installation.

Three components are identified in an electrical energy measurement system, such as the instrument transformers, the processing module, and the cloud computing system. Next, each component is described in detail.

### 3.1. Processing Module

The proposed IoT device uses the Photon Particle^®^ [29] to collect sensor data because the large number of analog inputs it handles, as well as its 12-bit resolution allows high signal sensitivity. Additionally, this technology has a direct connection to WiFi. This board allows publishing measurements in the free version with a latency of one second. Figure 4 illustrates the physical structure of this device.

### 3.2. Current and Voltage Sensors

The fundamentals parameters of electricity are reduced to a low-level voltage to be properly acquired by the Photon. The sensors that have been used for this purpose are the non-invasive current YHDC^®^ SCT-013 of 30 A [30], and the Voltage transformer of the Zeming^®^ brand, model ZMPT101B 2 mA [31]. Figure 5 shows both sensors.

The voltage signals provided by the sensors are properly modified to be acquired by the Photon. For this purpose, the signals from both sensors are coupled to the Photon through the circuits shown in Figure 6.

### 3.3. Integrated System

The previously presented devices were integrated in a Printed Circuit Board (PCB) shown in Figure 7.

Finally, all the components were mounted and housed in a plastic cabinet, with access to the microUSB Photon terminal for its programming, as well as attachments to place the designed device in any domestic load center. The image in Figure 8 shows the measuring device built and that in this work will be referred to as “IoTBEMS”, by their initials (IoT-Based Energy Metering System).

It is observed that the IoTBEMS is an extremely compact device and is enabled to be connected by anyone to a domestic installation. Internally it is protected against overvoltage in the electrical installation. The main features of the IoTBEMS are summarized in Table 5.

On the other hand, Table 6 summaries the costs of the components and materials used to develop this prototype. It shows that its cost does not exceed $46 USD, which is cheaper than the commercial devices.

### 3.4. Calibration

In the first instance, the board reads the voltage and current signals, and the electrical parameters are computed. For this purpose, we implemented the well-known relations for the computation of the true RMS values [32] such as: the active power and the energy consumption in kWh, whose expressions are given below.
(1)$$V_{trms}=\sqrt{\frac{v_{1}^{2}+v_{2}^{2}+\ldots +v_{N}^{2}}{N}}\;V$$
(2)$$I_{trms}=\sqrt{\frac{i_{1}^{2}+i_{2}^{2}+\ldots +i_{N}^{2}}{N}}\;A$$
(3)$$P=\frac{1}{1000N}\sum v_{k}i_{k}\;\mathrm{kW}$$
(4)$$kWh=\frac{1}{3600}\sum \frac{P}{f_{s}}$$
where *v_k_* and *i_k_* are the samples of voltage and current, respectively. In this case, we acquired *N* = 1000 samples at a frequency of *f_s_* = 1 kHz.

The behavior of the algorithm for the IoTBEMS was compared with a Power Quality Meter Fluke^®^ 435-II [33]. Both meters were mounted to measure the fundamental parameters of energy in a single phase of a conventional air conditioning system. Figure 9 shows this comparison for the power, current and voltage, respectively.

Note that all measured variables differ from that obtained with the 435-II. These differences are caused by the accuracy of the sensors and the latency of the internet connection. However, it can be concluded that the differences are not significant for the estimation of energy consumed.

### 3.5. Forecast of Energy Consumption

The IoTBEMS system was habilitated with an algorithm for forecasting the energy consumption, thus adding prognosis and diagnosis capabilities to the designed IoT system. The expression implemented uses the Statistical Mean (SM) of the kWh per each day elapsed. The following equation was implemented
(5)$$kWh_{a}=\frac{1}{E_{d}}\overset{E_{d}}{\underset{j=1}{\sum }}kWh_{j}$$
where *kWh_a_* is the kWh average over elapsed days, *kWh_j_* is the kWh per day, $E_{d}$ are the elapsed days. This expression is updated each day and is used to obtain early diagnosis on facilities conditions. To carry out the forecast of energy consumption, let us assume that user hold a regular domestic energy consumption. Based on this consideration, a simple forecast energy consumption is proposed
(6)$$kWh_{f}\approx D_{p}*kWh_{a}$$
where *D_p_* are the days per period to be forecast (30 or 60) that weigh the $kWh_{a}$ defined in (5). It is worth noticing, that high values in the first days can be attributable to failures or waste of energy.

## 4. The Proposed Interface

The interface for displaying the screens previously defined in Section 2, was designed using three programming environments, *html css*, and *js*. Figure 10 depicts a schematic diagram of the libraries employed for the interface designing; these libraries are available at [34].

Using the libraries previously mentioned, an appropriate interface was created. Figure 11 shows this interface along with the different screens of the previously defined conditions for each variable of interest. The reader can access through the hyperlink given in [35].

Figure 11a presents a menu where the user can navigate in the different screens. In the screen of the Figure 11b, the users can provide their personal data (optional) along with their cellphone and email to receive the information of the bill and the different alerts, as previously discussed in Section 2. Furthermore, in this screen the period of consumption can be selected. In the Figure 11c, a summary of the bill is displayed in real-time, which is indicated with the horizontal bars of the energy consumption for the different blocks according to the tariff selected. The behavior consumption through the days of the week using gradient colored bars is shown in Figure 11d. This gradient allows the user to determine if the consumption per day is higher than a limit previously established. Figure 11e uses the same idea but for the consumption by period. The forecasting of the energy is presented in the screen of Figure 11f, where consumption is obtained by extrapolation of the mean consumption of the previous days. The user can view announcements in real time as that showed in the screen of the Figure 11g, where a set of icons that change of color permits determining the level of consumption that has been reached. They also indicate if the user has a correct installation, or if the installation of PV systems needs a revision. Finally, the screen of the Figure 11h shows the information generated in the official printed bill version.

## 5. Experimental Results and Discussion

Finally, the developed system was applied in a real situation. It was mounted in a domestic installation in the state of Colima in Mexico during the period of August 19 to October 19 of this year. Figure 12 shows the image of the conventional bill, and the different screens of the interface show the measurement for this period, as well as the bill forecast obtained in the first week of measurement, and the announcement generated.

Figure 12 informs that the proposed system gives a measurement of 561 kWh (Figure 12b) vs. 579 kWh reported by the company (Figure 12a), this represents a minor difference of 18 kWh. The difference observed in the amount is attributed mainly to the block ranges considered in this study. Note that block B is two times higher than that used in the interface. On the other hand, the block IL is lower than used in the interface by 25 kWh, thus the significant difference for block S (129 vs. 211 kWh). According to the position of the pointer on the heat bar color, the householder under study is in the range of moderate high consumption since the Max allowed is 800 kWh.

Daily consumption is shown in Figure 12c, and it represents an attractive alternative for the users in the evaluation of their consumption day by day. The gradient colored bars help to rapidly determine if the consumption in a determined day if near to the mean allowed. In this case, all days present a similar level of consumption due to the actual pandemic situation of COVID-19, that has caused that people stay at home most of the time.

The forecast of the total energy consumption is shown in Figure 12d, and it was obtained with the first week of measurement. From this Figure, it is be concluded that the implemented algorithm estimates properly the final reading with a minor difference of 36 kWh (just 0.6 kWh per day), as appreciated in Figure 12b. Finally, this forecast was used for diagnosis of the installation, and the Figure 12e indicates in a friendly manner that the facilities are working properly, and changing the technology of lightning or the inverter-based fridges can reduce the energy consumption.

## 6. Conclusions

This work presented an application of IoT technology for obtaining a real time electricity bill for domestic applications. The presented interface is flexible since it implements all the tariffs, and it can be easily used in the Mexican country. An additional advantage of the proposed application is that it can be easily implemented in other countries with block-rate tariffs since it uses a simple programming algorithm. The proposed interface represents the first tool for diagnosis and prognosis for the Mexican users since the consumption can be forecasted with one week of measurements. Additionally, the proposed IoT system represents an alternative for identification of failures inside the installation. The IoT is a low-cost alternative for determining the energy consumption in real time, thus helping the users to save electric energy. The prognosis and diagnosis methods are simple, but the proposed tool is ready to include artificial intelligence methods. Additionally, this work will be extended in the future for three-phase applications in commercial and industrial systems including alerts for the user such as emails and cell text messages.

## Figures and Tables

**Figure 1 sensors-20-06178-f001:**
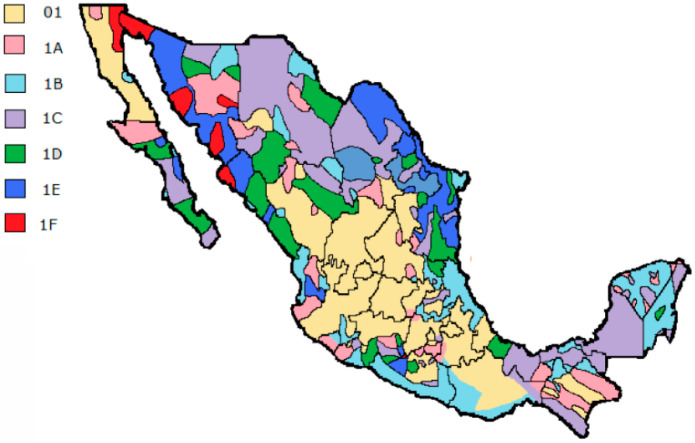
Geographic distribution of tariffs in Mexico [28].

**Figure 2 sensors-20-06178-f002:**
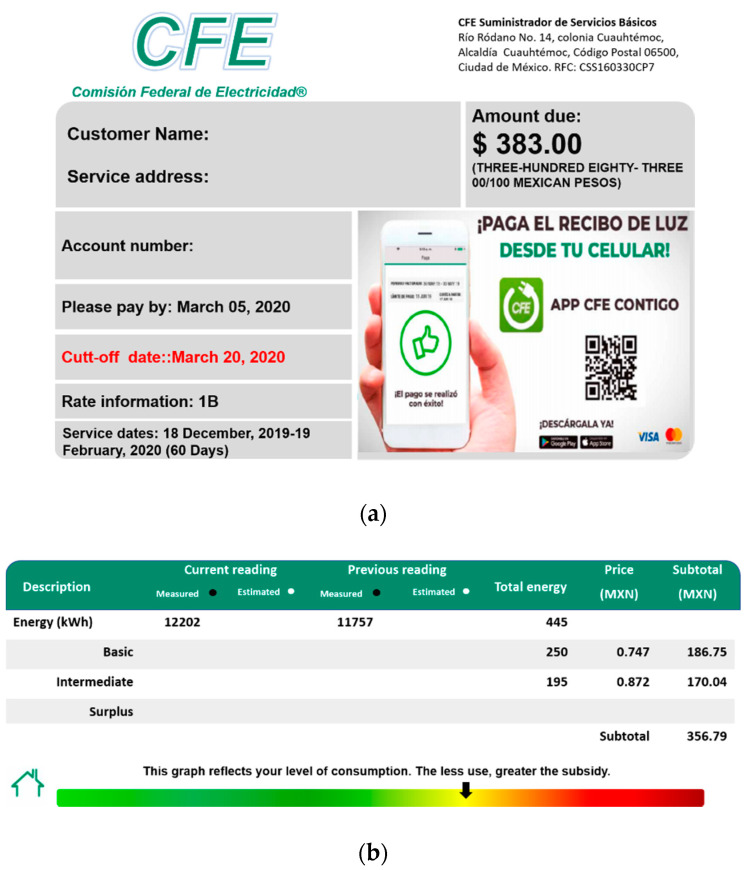
Sample of the Comision Federal de Electricidad (CFE) bill: (**a**) Amount due, period, and general information; (**b**) Distribution of kWh consumed in each level; (**c**) Details of taxes and other costs.

**Figure 3 sensors-20-06178-f003:**
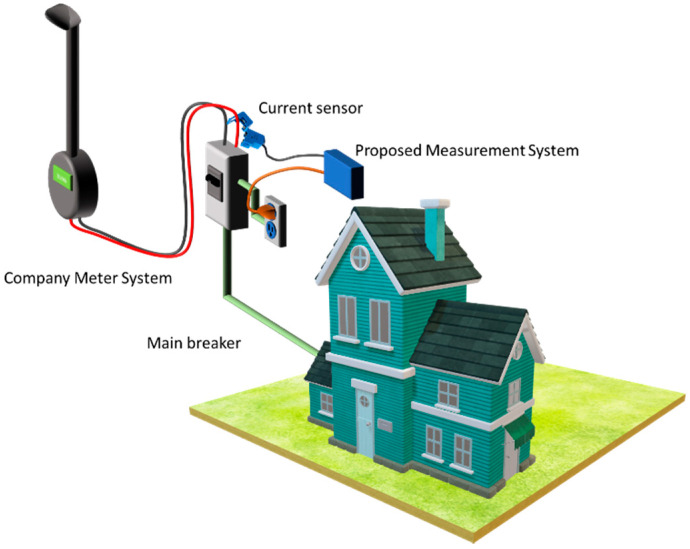
Schematic of the proposed measurement system.

**Figure 4 sensors-20-06178-f004:**
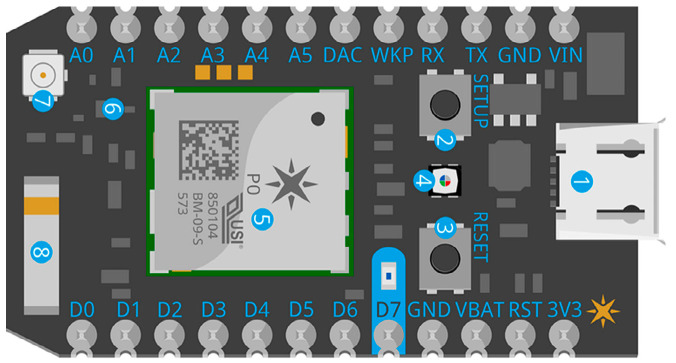
Physical structure of the Photon Particle^®^.

**Figure 5 sensors-20-06178-f005:**
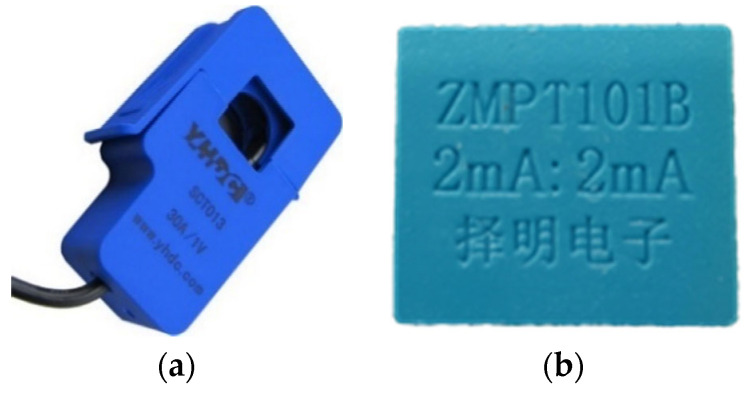
Current and Voltage sensors: (**a**) Current sensor, (**b**) Voltage sensor.

**Figure 6 sensors-20-06178-f006:**
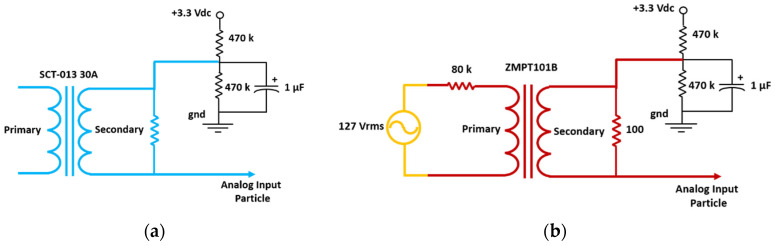
Coupling circuits based on instrument transformers: (**a**) Coupling circuit for current sensor, (**b**) Coupling circuit for voltage sensor.

**Figure 7 sensors-20-06178-f007:**
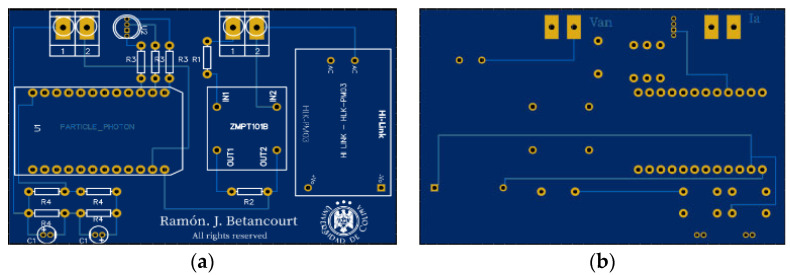
PCB board of the proposed meter system: (**a**) Top view, (**b**) Bottom view.

**Figure 8 sensors-20-06178-f008:**
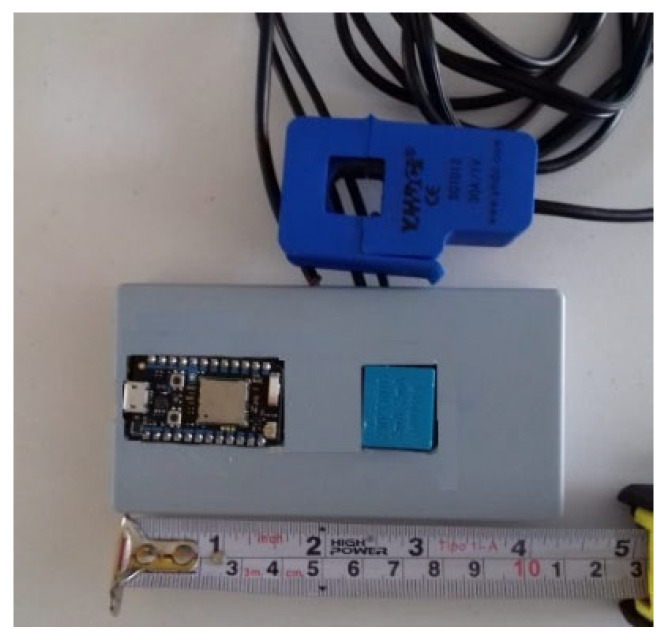
Photography of the IoTBEMS.

**Figure 9 sensors-20-06178-f009:**
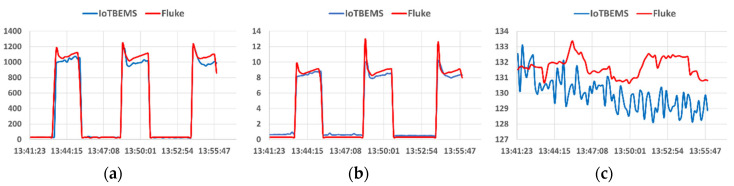
Comparison with Fluke 435 II: (**a**) Watts, (**b**) Amperes, (**c**) Volts.

**Figure 10 sensors-20-06178-f010:**
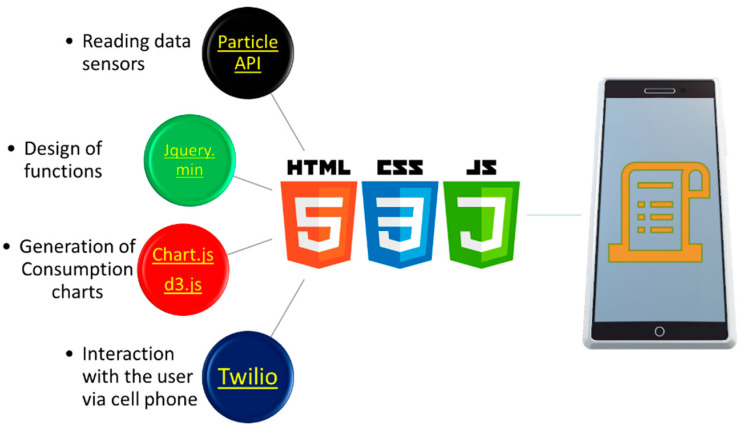
Schematic diagram for interface design.

**Figure 11 sensors-20-06178-f011:**
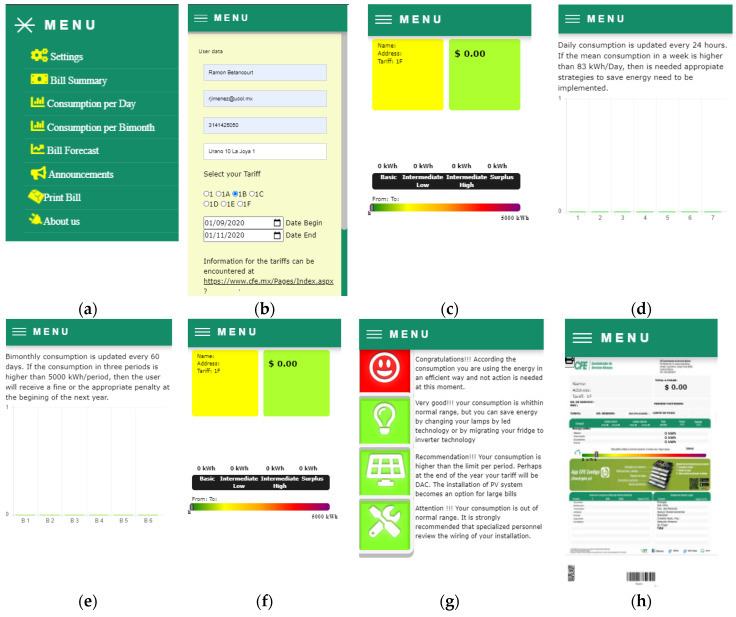
Screens for the user interface: (**a**) Menu, (**b**) Data and Tariff user, (**c**) Summary of consumption (**d**) Daily consumption, (**e**) Bimonthly consumption, (**f**) Bill forecast, (**g**) Announcements, (**h**) Print bill.

**Figure 12 sensors-20-06178-f012:**
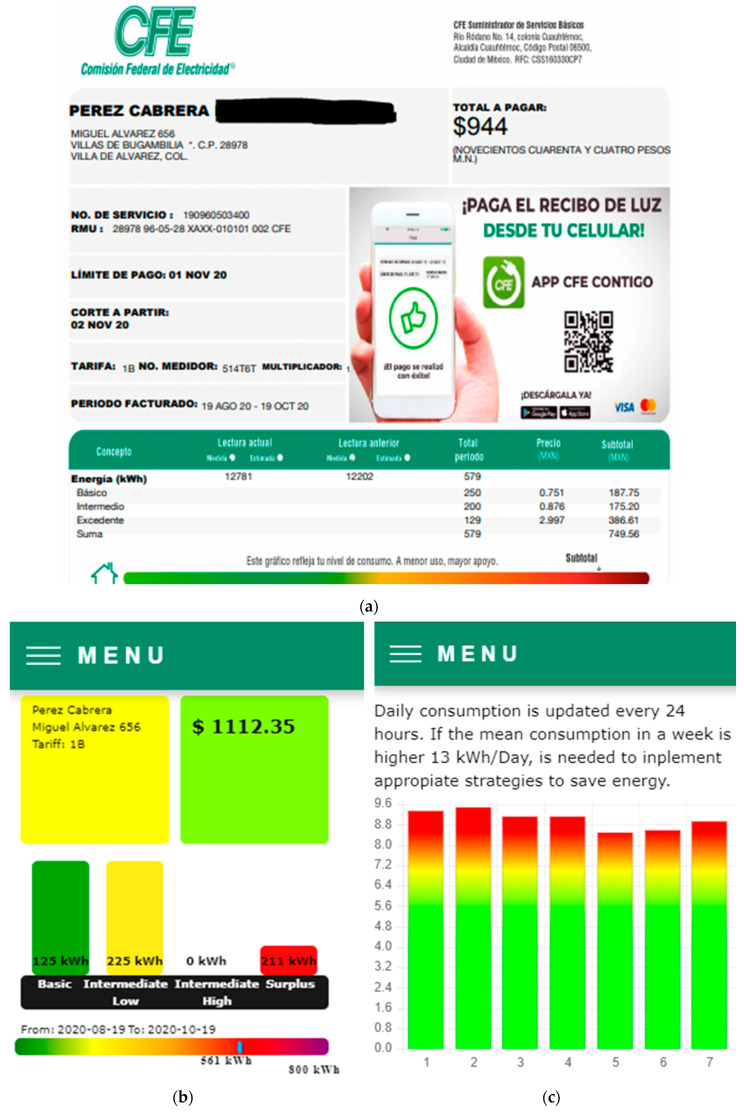
Summary of bills: (**a**) Receipt of the bill company (**b**) Bill summary (**c**) Bill forecast for first week (**d**) Daily consumption (**e**) Announcements.

**Table 1 sensors-20-06178-t001:** Commercial home monitoring systems.

Brand Name	Identify Appliances	Support for PV System	Approximated Price (USD)	Web Page
Sense Energy Monitor	✓	✓	$300.00	https://sense.com/
Neurio Home Energy Monitor	✓	✘	$45.00	https://www.neur.io/
Eyedro EHEM1	✘	✘	$119.00	http://eyedro.com/home-electricity-monitors/
TED Pro Home Electricity Monitor	✘	✓	$299.95	http://www.theenergydetective.com/tedprohome.html
Energy Engage Hub	✘	✓	$139.99	https://engage.efergy.com/

**Table 2 sensors-20-06178-t002:** Summary of Domestic Tariffs in Mexico.

Tariff	Block				Max ^5^	**T °C**
	B ^1^	IL ^2^	IH ^3^	S ^4^		
**1**	**0→75**	75→140	-	140→	3000	>25
1A	0→100	100→150	-	150→	3600	25
1B	0→125	125→225	-	225→	4800	28
1C	0→150	150→300	300→450	450→	10,200	30
1D	0→175	175→400	400→600	600→	12,000	31
1E	0→300	300→750	750→900	900→	24,000	32
1F	0→300	300→1200	1200→2500	2500→	30,000	33

^1^ Basic, ^2^ Intermediate low, ^3^ Intermediate high, ^4^ Surplus, ^5^ Maximum kWh allowed per year.

**Table 3 sensors-20-06178-t003:** Screens of the interface and their objective.

Screen	Objective
1	General data of the user, selection of tariff, begin and end of reading period.
2	Summary of consumption on real time.
3	Consumption for each day in a week.
4	Consumption for each period in a year.
5	Forecast of the consumption obtained with the mean of consumption.
6	Announcements when the consumption is within the moderate or excessive range.
7	Displaying real-time consumption on the company format.

**Table 4 sensors-20-06178-t004:** Strategies of visualization and alerts for the variables.

Variable	Visualization Method	Alert Type	When
kWh in the period	Gradient color progress bar	SMS	It is higher than Max/6 or Max/12
kWh Basic	Green progress bar	None	None
kWh Intermediate Low	Yellow progress bar	None	None
kWh Intermediate High	Orange progress bar	None	None
kWh Surplus	Red progress bar	SMS	It is reached before the middle the period
kWh accumulated over year	Numeric	SMS	It is near to the Max
Total and Pay due	Alphanumeric	SMS	five days before the deadline in each period
Daily consumption	Chart bar with gradient color	None	None
Bimonthly (monthly) consumption	Chart bar with gradient color	None	None
Diagnosis of consumption	Animated icon	None	None

**Table 5 sensors-20-06178-t005:** Main characteristics of IoTBEMS.

IoTBEMS	
Maximum current	30 A
Maximum voltage	150 V
Number of additional current sensors that can be added	5
Number of digital channels	8
Open architecture	Yes

**Table 6 sensors-20-06178-t006:** IoTBEMS Budget.

Device	Cost (USD)
Particle Photon	$20.00
Sensor SCT-013 30A	$8.00
Sensor ZMPT101B	$8.00
PCB	$3.00
Box container	$2.00
Miscellaneous components	$5.00
Total	$46.00

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
