# Peer review of "IoT-Based Electricity Bill for Domestic Applications"

_sensors, 2020, doi:10.3390/s20216178_

Round 1

Reviewer 1 Report

The authors in this paper presented the application of IoT in obtaining a real time electricity bill for domestic applications by considering safety in a domestic installation.

The processing module is Photon Particle and various sensors e.g. current and voltage sensors are used and the signals from the both sensors are coupled to the Photon through the coupling circuits which are integrated into a PCB board and an IoT-Based Energy Metering System namely IoTBEMS is resulted. A comparison of the proposed system against a commercial power quality meter Fluke is presented as well. This can be considered as the most technical contribution of the paper however, the technical contribution looks limited. Please provide justification what is technical contribution of this research work comparing to the most recent researches and also indicate on the novelty of the approach. For example in a relevant publication in the field i.e., a novel smart ECO model for energy consumption optimization considers the energy efficiency and consumptions in the home scenarios, which uses similar sensors as well as some AI algorithms. This is not attempting to have a billing system but it can also be achieved with similar system. Please elaborate further difference between this and your work. 

The authors indicated that “Some prototypes present some advantages over others, such the presented in [13], where and Arduino Uno Wi-Fi Rev v2 is used”, however Arduino Uno is not a powerful processing unit in particular for commercial systems. Researchers and companies usually use more powerful boards. Please try to revise the Introduction section with more recent progress in the field.

Can figure 2 be translated in English and an English version be added instead?

In line 254, is it Fluke PQ43 or Fluke 435?

Almost half of the references are online references, which needs to be revised with adding most recent research scientific articles in the reference list.

Overall, the authors are addressing a very interesting topic, which can be beneficial in various countries however, more justification on technical contribution and novelty needs to be added as well as more scientific references.

Author Response

Thanks to the Reviewer for their suggestions to improving the quality of the manuscript. In this new version of the paper all the recommendations were addressed.

Reviewer 2 Report

The paper presents an IoT system to define the real-time electriity bill based on the energy used in domestic facilities in Mexico.

The novelty of the paper in confront of the state of the art is not well explained. The authors have to identify what are the innovative components in the paper and explain it.

Furthermore, the results are not so relevant in the way they were presented. The authors can show the results during the all period of monitoring and not just the final results.

The authors have also to explain how they defined the accuracy between the IoT BEMS and the Power Quality meter Fluke presenting the results in the measurement of accuracy and the method that they used.

The conclusions of the paper have to be supported by the results.

Author Response

Thanks to the Reviewer, we appreciate their comments and suggestions. All observations were attended.

Round 2

Reviewer 1 Report

One of my previous major comments was doing further research on powerful processing board than Arduino Uno, which the authors addressed it by revising the Introduction section.

My comment on the contribution is also addressed. Furthermore, the correction on FLUKE 435 was also done properly.

There is a very minor comment on "Domestic" term used in the paper, what this exactly means? Does it mean "household" as most of the references in this paper indicates too? Make sure a right term is used.

Reviewer 2 Report

The authors clearly replied to my comments.